# Digital Light Processing 3D Printing of Gyroid Scaffold with Isosorbide-Based Photopolymer for Bone Tissue Engineering

**DOI:** 10.3390/biom12111692

**Published:** 2022-11-15

**Authors:** Fiona Verisqa, Jae-Ryung Cha, Linh Nguyen, Hae-Won Kim, Jonathan C. Knowles

**Affiliations:** 1Division of Biomaterials and Tissue Engineering, Eastman Dental Institute, University College London, London NW3 2PF, UK; 2Department of Chemistry, Dankook University, Cheonan 31116, Republic of Korea; 3UCL Eastman-Korea Dental Medicine Innovation Centre, Dankook University, Cheonan 31116, Republic of Korea; 4Institute of Tissue Regeneration Engineering (ITREN), Dankook University, Cheonan 31116, Republic of Korea; 5Department of Nanobiomedical Science and BK21 PLUS NBM Global Research Center for Regenerative Medicine, Dankook University, Cheonan 31116, Republic of Korea

**Keywords:** 3D printing, photopolymer, digital light processing, bone regeneration

## Abstract

As one of the most transplanted tissues of the human body, bone has varying architectures, depending on its anatomical location. Therefore, bone defects ideally require bone substitutes with a similar structure and adequate strength comparable to native bones. Light-based three-dimensional (3D) printing methods allow the fabrication of biomimetic scaffolds with high resolution and mechanical properties that exceed the result of commonly used extrusion-based printing. Digital light processing (DLP) is known for its faster and more accurate printing than other 3D printing approaches. However, the development of biocompatible resins for light-based 3D printing is not as rapid as that of bio-inks for extrusion-based printing. In this study, we developed CSMA-2, a photopolymer based on Isosorbide, a renewable sugar derivative monomer. The CSMA-2 showed suitable rheological properties for DLP printing. Gyroid scaffolds with high resolution were successfully printed. The 3D-printed scaffolds also had a compressive modulus within the range of a human cancellous bone modulus. Human adipose-derived stem cells remained viable for up to 21 days of incubation on the scaffolds. A calcium deposition from the cells was also found on the scaffolds. The stem cells expressed osteogenic markers such as RUNX2, OCN, and OPN. These results indicated that the scaffolds supported the osteogenic differentiation of the progenitor cells. In summary, CSMA-2 is a promising material for 3D printing techniques with high resolution that allow the fabrication of complex biomimetic scaffolds for bone regeneration.

## 1. Introduction

Bone is one of the most transplanted human body tissues, with an autogenous bone graft as the gold standard [1,2]. However, the harvesting process of autologous bone grafting has been reported to cause donor site pain and infection, increased blood loss, prolonged surgery duration, and hospitalisation [3]. The graft also has a limited supply, since it is harvested from the same patient to reduce the possibility of graft rejection, which is one of the risks of all grafts. Synthetic bone grafts have been developed as an alternative to these grafts. Calcium-based bone substitutes are the most used synthetic products, particularly in powder or granule form. This type of synthetic graft is not suitable for the management of large bone defects. A critical size defect requires a strong graft that allows both osteogenesis and angiogenesis to prevent necrosis and implant failure, due to the loading condition that the human body endures [4]. Inducing osteogenesis can be done by incorporating cells or growth factors into the implants, whilst creating pores on the graft will help to induce vascularisation [5]. This approach combines reconstructive surgery and tissue engineering to restore bone defects.

The use of 3D printing or additive manufacturing has emerged as a promising method for tissue engineering. One of the advantages of 3D printing is the possibility of fabricating defect-specific scaffolds or patient-specific implants, based on computed tomography data that are translated into computer-aided designs (CAD) [6]. The most popular 3D printing method for tissue engineering nowadays is extrusion-based printing [7]. Extrusion-based 3D printing extrudes material from the printer’s nozzle, and then the extruded materials undergo a light-curing or cross-linking process to establish the 3D construct. Hydrogels are the common material for this type of printing since they have suitable rheological properties for the extrusion method. This method also enables cells to be incorporated into the hydrogels. The temperature, pressure, and speed settings can be adjusted to ensure cell viability. However, gels do not have adequate mechanical properties for hard tissue engineering, which requires a scaffold with mechanical properties that can withstand load-bearing situations and surgical procedures [8]. The 3D printing designs are also limited, due to the extrusion mechanism and nozzle diameter, which do not allow the fabrication of interlacing structures [9].

Another type of 3D printing that has the potential for bone tissue engineering is light-based 3D printing. This method exposes the liquid photopolymer to a UV light beam that solidifies the polymer through polymerisation. The printing techniques that use this mechanism are stereolithography (SLA) and digital light processing (DLP) (Figure 1). Light-based 3D printing methods are also known for their high accuracy, precision, and faster printing speed. SLA uses a laser beam, whilst DLP has a projector to solidify the resin layer-by-layer and create a solid 3D construct. SLA can build structures with larger volumes, but DLP offers faster printing speed with high resolution. The current maximum resolution of DLP and SLA is reported to be within 25–50 μm [10]. DLP printers are also cheaper than SLA printers.

Pores on the scaffold are found to have an important role in bone tissue engineering and bone regeneration. The suitable pore size allows nutrient and metabolite transport and supports cell proliferation. The range of favourable pore size diameters for those purposes is 100–400 μm [12]. This range corresponds to the cancellous bone structure of the human bone (Figure 2). The interconnectivity of the pores is also important for cell migration and maximising nutrient diffusion. A structure with interconnected micropores can efficiently be designed using 3D printing, especially with DLP, which can print at high resolution and complex designs.

The challenge with the DLP method is to find a photopolymer that is suitable for the printing mechanism and is biocompatible. The widely available commercial photopolymer resins are toxic and unsuitable as biological implants for the human body. CSMA-2 is a novel, isosorbide-based polymer that shows excellent biocompatibility in vitro and in vivo, as well as excellent printability with light-based 3D printing [14,15]. Isosorbide is a D-sorbitol derivative demonstrating promising mechanical properties due to its bicyclic structure [16]. Since isosorbide derives from sugar, it counts as a renewable and sustainable bio-based compound [17]. It is inexpensive, non-toxic, and has been incorporated into materials such as polycarbonates, polyamides, and polyurethane via step-growth polymerisation [18]. Good optical clarity makes isosorbide suitable as a monomer for a 3D printing photopolymer [18,19]. Light-cured, isosorbide-based CSMA-2 has been reported to have mechanical properties similar to human cancellous bone and was non-toxic to MG63 cell lines [14,15]. Solid disc and log pile structures were successfully printed accurately using CSMA-2 as the 3D printing material in the previous study [15]. Therefore, this study aimed to characterise and utilise CSMA-2 as the 3D printing material to fabricate a biocompatible and strong 3D construct with high 3D printing resolution and a complex triply periodic minimal surface or gyroid CAD by using the DLP printing technique for bone tissue engineering.

## 2. Materials and Methods

### 2.1. Materials

Isosorbide, ethylene carbonate, IPDI (Isophorone diisocyanate), TEGDMA (Triethylene glycol dimethacrylate), DBTBL (dibutyltin dilaurate), HEMA (2-hydroxyethyl methacrylate), penicillin/streptomycin, and L-glutamine were purchased from Sigma Aldrich (Darmstadt, Germany). α modified Eagle’s medium (α MEM), fetal bovine serum (FBS), and MesenPro medium were obtained from Gibco, Life Technologies Ltd., Paisley, UK.

### 2.2. CSMA-2 Synthesis

The CSMA-2 synthesis was done by following previous methods (Figure 3) [14,15]. The synthesis was started with the synthesis of BHIS by reacting isosorbide (100 g, 684.3 mmol) and ethylene carbonate (132.57 g, 1505.5 mmol) that were degassed under dry nitrogen for 60 min. The reaction was then heated on a hot plate for one hour at 70 °C. After the solid components were completely dissolved, the reaction mixture was heated to 170 °C. Then, potassium carbonate (3.0 g, 21.71 mmol) was added, and the mixture was left to react for 48 h. The resulting BHIS was purified through silica column chromatography using methanol and ethyl acetate (1:9). The purified BHIS was then evaporated in a rotary evaporator to remove the solvents.

The next step was reacting the purified BHIS (32.15 g, 79.37 mmol) with IPDI (57.15 g, 257.07 mmol), TEGDMA (125 g, 436.56 mmol), and 5 drops (approximately 0.5 mL) of DBTDL at 25 °C for 4 h. After that, HEMA (71.42 g, 548.82 mmol) and another 5 drops (approximately 0.5 mL) of DBTDL were added into the reaction mixture and left to react for 12 h at 25 °C, resulting in the final CSMA-2 monomer ((3R, 3aR, 6S, 6aR)-hexahydrofuro [3,2-b] furan-3,6-diyl)bis(oxy)) bis(ethane-2,1-diyl))bis(oxy))bis(carbonyl))bis(azan ediyl))bis(3,3,5-trimethylcyclohexane-5,1-diyl))bis (azanediyl))bis(carbonyl))bis(oxy))bis(ethane-2,1-diyl) bis(2-methylacrylate)).

Phenylbis (2,4,6-trimethylbenzoyl) phosphine oxide, or BAPO, (Sigma Aldrich) was used as the photoinitiator. A 2 wt% of BAPO was added to the CSMA-2 and left to stir for 24 h. Hydroxyapatite or HA (Captal R, Plasma Biotal, UK) with a 1.67 Ca:P ratio and particle size ranging from 6–20 μm, was added and mixed into the CSMA-2 using a speed mixer at 1700 RPM for 2 min. The HA addition to the CSMA-2 was 5% wt and 10% wt. The final CSMA-2 groups were CSMA-2 0HA (without HA), CSMA-2 5HA (5% HA), and CSMA-2 10HA (10% HA).

### 2.3. CSMA-2 Monomer Characterisation

#### 2.3.1. Degree of Conversion

Fourier-transform infrared spectroscopy (FTIR, System 2000, PerkinElmer, Seer Green, UK) was used to determine the monomer degree of conversion. The monomer was dropped on the diamond of an attenuated total reflectance accessory (Golden Gate ATR, Specac Ltd., Orpington, UK) and exposed to a Demi Plus LED light-curing unit for 20 min at 20 °C. The spectra were then recorded to analyse the conversion. The absorbance profiles were measured at 1319 ± 1 cm^−1^ (C–O stretch bond) and 1334 ± 2 cm^−1^ (baseline). The conversion was calculated by using the following.
$$C=\left[1-\left(\frac{Af}{A0}\right)\right]\times 100$$

*C* is the conversion; *Af* is the final absorbance; and *A*0 is the initial absorbance.

#### 2.3.2. Rheology

The rheological properties for optimising the CSMA-2 formulations were analysed using HAAKE Viscotester iQ Rheometers (Thermo Scientific, Walthman, MA, USA). A rotational shear test with a controlled shear stress from 1 to 1000 Pa was performed at 20 °C for 300 s. The data were analysed with HAAKE RheoWin software (Thermo Scientific, Walthman, MA, USA).

#### 2.3.3. 3D Printing

The solid and gyroid constructs were fabricated using a Nobel Superfine DLP 3D printer (XYZ Printing, New Taipei City, Taiwan) (Figure 4). Based on the existing repositories, the constructs were designed with computer-aided design (CAD) software (Meshmixer, Autodesk, San Francisco, CA, USA). A slicing software (XYZware Nobel, XYZ Printing, Taiwan) was used to slice the design and determine the printing setup. For the base setup, the curing time was 19 s with 60 W/m^2^ power intensity. The curing time for the intermediate and model setups was 8.3 s with 53 W/m^2^ power intensity. All the setups used 15% of the power level and 0.25 mm/s for the speed at 20 °C. After the printing was finished, the samples were washed with 99% methanol (Merck, Kenilworth, NJ, USA) for 5–10 min to remove the uncured monomer, then left to dry, followed by a post-curing process with a UV chamber (XYZ Printing, New Taipei City, Taiwan) for 1 min at level 3 intensity.

### 2.4. 3D-Printed Scaffold Characterisation

#### 2.4.1. Printing Resolution and Scaffold Morphology

Scanning electron microscopy (SEM) (Philips XL30 field emission SEM, Amsterdam, The Netherlands) was used to evaluate the printing resolution and the scaffold morphology. Before the analysis, the samples were coated with 95% gold and 5% palladium (Polaron E5000 Sputter Coater, Quorum Technologies, Laughton, UK). The printing resolution was observed by measuring the layer thickness. The printing resolution was set to 0.1 mm.

#### 2.4.2. Wettability

The wettability of the 3D-printed scaffold was examined by calculating the surface energy of the 3D-printed flat sample surface. The contact angles of the water, glycerol, and di-iodomethane were obtained using a KSV instruments Cam 200 optical contact angle meter (Biolin Scientific, Manchester, UK).

#### 2.4.3. Mechanical Properties

A compressive test was performed using Shimadzu Autograph AGS-X machinery (Shimadzu, Milton Keynes, UK). Gyroid cylinders with six repetitions were used as samples. The preload was performed at 3 mm/min speed with a maximum force of 1 N. The cylinders were compressed at a crosshead speed of 1 mm/min until the sample failed. The data were obtained via TRIOS software (TA Instruments, Crawley, UK).

### 2.5. 3D-Printed Scaffold In Vitro Studies

#### 2.5.1. 3D Cell Culture

Human Adipose-Derived Stem Cells (hADSC) were obtained from Lonza and cultured with MesenPRO medium (Gibco), 1% penicillin/streptomycin (P/S) (Sigma-Aldrich, Darmstadt, Germany), and 1% L-glutamine (Sigma-Aldrich). The cells were incubated at 37 °C and 5% CO_2_.

The 3D-printed samples with the gyroid structure were sterilised with 70% alcohol for 15 min, washed with PBS twice, and then left to dry. UV light sterilisation was then performed for 15 min on each side. The samples were soaked with a complete medium and placed in the incubator for 24 h before the cell seeding. After removing the medium, passage 5 cells were seeded to the scaffold surface and incubated for 1 h. Fresh medium was added afterwards. The cell density was 5 × 10^4^ per scaffold. The medium was changed every 2–3 days.

#### 2.5.2. Metabolic Activity

To analyse the metabolic activity, 10% (*v*/*v*) alamarBlue (Invitrogen, Thermo Fisher, Walthman, MA, USA) was added to each well and incubated for 4 h at 37 °C. A Biotek FLx800 microplate reader was used to read the fluorescence intensity with 540/35 and 600/40 excitation/emission wavelengths. Four samples were prepared for each scaffold group. The scaffolds were incubated for 21 days.

#### 2.5.3. Cell Attachment

The cell attachment was analysed by observing the hADSC incubated on the 3D-printed scaffolds with SEM (Zeiss Sigma, Oberkochen, Germany). The scaffolds with cells were fixed in a 3% glutaraldehyde and 0.1 M cacodylate buffer and then kept at 4 °C for 24 h. The samples underwent serial ethyl alcohol dehydration and critical drying with hexamethyldisilazane. The samples were coated with 95% gold and 5% palladium afterwards (Polaron E5000 Sputter Coater, Quorum Technologies, Laughton, UK).

#### 2.5.4. Osteogenic Differentiation

To induce the osteogenic differentiation, the hADSC were cultured with an osteogenic medium (Mesenchymal Stem Cell Osteogenic Differentiation Medium, PromoCell, Germany) after the cell seeding. The medium was changed every three days.

#### 2.5.5. Calcium Deposit

Alizarin red staining (ARS) (Sigma Aldrich) was performed to evaluate the calcium deposit of the hADSC cultured in the 3D-printed scaffold. The staining was carried out on days 7, 14, and 21. The medium was removed from the samples and washed using PBS three times. The Alizarin red staining solution was added to the scaffold samples and incubated for 5 min at room temperature. The staining was then removed by washing the scaffold using PBS. The stained scaffolds were photographed using a Canon EOS camera.

#### 2.5.6. Protein Expression

Immunofluorescence was performed to observe the protein expression of the hADSC. The RUNX2 (Runt-related transcription factor 2), OCN (osteocalcin), and OPN (osteopontin) expressions were observed as markers of osteogenic differentiation on day 7, day 14, and day 21. The samples were fixed with 4% paraformaldehyde for 10 min, according to the time points, and then washed three times with ice-cold PBS. The samples were incubated for 10 min with 0.1% Triton X and then washed with PBS three times for 5 min. To block the unspecific binding of the antibodies, the samples were incubated with 1% BSA for 30 min. The primary antibody incubation of the RUNX2 (1:200) (Ab192256, Abcam, Cambridge, UK), OCN (1:100) (MAB1419, Novus Biological, Littleton, CO, USA), and OPN (1:200) (ab8448, Abcam, UK) were done overnight at 4 °C in a humidified chamber. The antibody solutions were then removed, and the samples were washed thrice with PBS, for 5 min each wash. The secondary antibodies AlexaFluor 488 (1:200) (Abcam, UK) and AlexaFluor 594 (1:200) (Abcam, UK) were added, and the samples were incubated for 1 h at room temperature in the dark. After that, the solution was removed, and the samples were rewashed three times with PBS for 5 min. For the counterstaining, the samples were incubated with DAPI (0.4 μg/mL) (Invitrogen) for 10 min and iFluor 647 (Abcam, UK) for 30 min, then washed with PBS. The images were collected using a confocal microscope (Aurox, Abingdon, UK) and Visionary software (Aurox, UK).

#### 2.5.7. Gene Expression

A gene expression assay was performed by isolating the RNA from the samples using Direct-zol RNA kits (Zymo Research, Irvine, CA, USA), according to the protocol, on day 7, day 14, and day 21. Three biological replicates and two technical replicates were used. The isolated RNA from the samples was converted to cDNA using a High-Capacity cDNA Reverse Transcription Kit (Applied Biosystems, Thermo Fisher Scientific, Walthman, MA, USA). A real-time quantitative polymerase chain reaction (RT-qPCR) was performed using the TaqMan Fast Advanced Master Mix (Applied Biosystems, Thermo Fisher Scientific, Walthman, MA, USA) and TaqMan gene expression assay (Applied Biosystems, Thermo Fisher Scientific, Walthman, MA, USA). The target genes were RUNX2 (Hs01047973_m1), SPP1 (osteopontin) (Hs00959010_m1), and GAPDH (Hs02786624_g1) as reference. The RT-qPCR was processed using the Applied Biosystems 7300 Real-Time PCR System (Thermo Fisher Scientific, Walthman, MA, USA). The CT value of each target gene was subtracted by the GAPDH CT values from the samples. The ΔCT of the sample group was then subtracted by the ΔCT of the hADSC seeded on the tissue culture plate with the osteogenic medium at the same time point to obtain the ΔΔCT. The final values were 2^−ΔΔCT^ or relative gene expression.

### 2.6. Statistical Analysis

The data were presented as a mean and standard variation or box plot. The statistical analyses were conducted using IBM SPSS statistics 28 (IBM, Armonk, NY, USA). We used one-way ANOVA with a Tukey post-test analysis.

## 3. Results

### 3.1. CSMA-2 Monomer Characterisation

Each CSMA-2 group demonstrated a similar conversion rate and was not significantly different from the mixture that had the hydroxyapatite, or not, after being exposed to the UV light. More than 50% of the monomer was polymerised, as can be seen in Figure 5a. The conversion rate for the CSMA-2 0HA, CSMA-2 5HA, and CSMA-2 10 HA were 62%, 56%, and 60%, respectively.

We then analysed the CSMA-2 rheological properties to determine its printability and 3D printing settings. As can be seen from Figure 5b, the shear stress (τ) of the CSMA-2, with and without the HA, is proportional to the shear strain (γ). This is a typical Newtonian material flow behaviour. The addition of HA increased the CSMA-2 viscosity but did not change its Newtonian properties, as confirmed by Figure 5c–e, since the viscosity (η) is constant throughout the different shear strains. The viscosity values were approximately 0.3, 0.4, and 0.5 Pa.s for the CSMA-2 0HA, CSMA-2 5HA, and CSMA-2 10 HA, respectively.

### 3.2. 3D Printing and Scaffold Characterisation

To evaluate the printing resolution, a pyramid structure was printed, and the layer thickness was measured. The printing resolution, or the distance between the layers, was set at 0.1 mm or 100 μm. Based on the SEM measurement, the 3D-printed resolutions for the CSMA-2 0HA, CSMA-2 5HA, and CSMA-2 10HA were 86, 83, and 84 μm, respectively (Figure 6). A complex gyroid structure with interconnected pores was successfully printed using DLP and CSMA-2 as the photopolymer. Figure 7 shows the CAD and 3D-printed scaffolds, with apparent similarities between the design and the 3D-printed construct in dimension and architecture. The colour of the 3D-printed scaffold could be described as ivory with different opacity among the groups. Although adding HA increased the mixture’s viscosity, the 3D printing process and result were not significantly affected. Macroscopically, the structure and size were not significantly different among the CSMA-2 groups. However, Figure 8a–c showed different surface morphology, as expected. The higher the HA content, the increased roughness of the surface. A relatively smooth surface can be observed on a 3D-printed CSMA-2 scaffold without the HA, whilst CSMA-2 5HA and 10HA showed rough and irregular surface morphology.

The water contact angle was lower on the 3D-printed scaffolds’ surface with the HA, whilst the surface energy was higher. The water contact angle was 76°, 74°, and 62° for the CSMA-2 0HA, CSMA-2 5HA, and CSMA-2 10HA, respectively (Figure 9a). The CSMA-2 0HA, CSMA-2 5HA, and CSMA-2 10HA had surface energy of 41, 47, and 53 mN/m, respectively (Figure 9b).

As can be seen from Figure 9c, the CSMA-2 10 HA showed the highest compressive modulus (0.54 N/mm^2^) among the group, followed by the CSMA-2 with 5HA (0.51 N/mm^2^), and the CSMA-2 0HA (0.43 N/mm^2^).

In general, the metabolic activities of the hADSC cells were higher when cultured with the growth medium than with the osteogenic medium (Figure 10). The CSMA-2 0HA showed the highest metabolic activity in the osteogenic and growth media, particularly on day 21. However, there were differences in the metabolic activity trends between the scaffold groups in the osteogenic and growth media. In the osteogenic medium, the hADSC metabolic activity peaked on day 7 and decreased on days 14 and 21, whilst the hADSC incubated in the growth medium continued to increase and peaked on day 21, except for the CSMA-2 5HA group, which showed the highest metabolic activity on day 14.

Figure 11 shows the hADSC attachment on the 3D-printed CSMA-2 scaffolds. The cells were found to attach and spread on the scaffolds. Calcium-phosphate nodules were also visible around the cells, including cells that were seeded on the CSMA-2 0HA scaffolds. The different surface morphology of the scaffold groups was also noticeable, with the CSMA-2 5HA and 10HA scaffolds showing rougher surface morphology than the CSMA-2 0HA groups.

The alizarin red staining images (Figure 12) show the different intensities of staining between the various media and scaffold groups. The scaffolds incubated in the osteogenic medium showed stronger positive staining than those in the growth medium. The staining was more intense on the scaffolds with the HA, although the CSMA-2 0HA scaffold incubated in the osteogenic medium already showed increasing intensity on day 14. The staining intensity also increased following the incubation period.

Figure 13, Figure 14 and Figure 15 show the osteogenic protein marker expression on the CSMA-2 scaffold. The RUNX2, OPN, and OCN expressions were detected from day 7 of incubation with the osteogenic medium. The expression of RUNX2 was relatively stronger on the CSMA-2 0HA scaffolds and showed no noticeable difference within the incubation period. The OPN expression was also observed on day 7 on all the scaffold groups and remained detected until day 21. Similar to the OPN, the OCN expression can be observed on day 7, with the strongest expression on day 14, and still can be seen on day 21.

The gene expressions of RUNX2 were not significantly different between the CSMA-2 0HA, 5HA, and 10HA (Figure 16a). The CSMA-2 0HA scaffolds showed significantly lower OPN gene expression on day 7 and day 21 compared to the CSMA-2 5HA scaffold group. As can be seen from Figure 16b, the OPN gene expression of the hADSC on the CSMA-2 5HA scaffold increased by twofold compared to the control at day 7. On day 14, the OPN gene expressions were not different among the scaffold groups.

## 4. Discussion

CSMA-2 was successfully synthesised by following the previous methods [14,15]. The final result of the synthesis was a clear, viscous mixture, which was expected from copolymerising Isosorbide [19]. This optically transparent mixture enables polymerisation via a light cure. The degree of conversion was also similar to the previous studies, with more than 50% of the monomer being polymerised after exposure to UV light for less than 1 min [14,15]. This result confirmed that, after the addition of BAPO as a photoinitiator, CSMA-2 could act as a photopolymer suitable for light-based 3D printing. The degree of conversion of the dimethacrylate monomer, one of the CSMA-2 components, is also reported to be between 55% and 75% after the irradiation [20]. It is common for methacrylate monomers to exhibit residual monomers. The factors that can influence the degree of conversion include the wavelength of the light source. We used an LED light-curing unit with 450–470 nm for the degree of conversion analysis with the FTIR. The photoinitiator used in this research was BAPO, which has light absorption ranging from 296 nm to 370 nm [21,22]. This might affect the degree of conversion, since BAPO is more suitable with a light source with a lower wavelength, such as a DLP printer with a light source wavelength of 405 nm, which is also why BAPO was chosen. In 3D printing, the residual monomer can be removed following the post-printing process, such as washing with alcohol and post-curing. This process will also improve the quality of the 3D-printed structures, particularly those with micropores.

Based on the rheological analysis, the CSMA-2, with or without the HA, was a Newtonian material. Its shear rate was proportional to its shear stress. Every CSMA-2 mixture group showed constant viscosity throughout different shear rates (Figure 5d–f). Different 3D printing methods require different printing materials with different rheological properties. DLP will be more suitable for Newtonian material, since it does not use pressure or extrusion as its printing mechanism. It can fabricate favourable architecture with high resolution that can support bone regeneration, such as interconnected pores with a 100 to 400 μm diameter, which allow bone ingrowth [12,23]. Bone architectures are also varying in different anatomical structures, e.g., jawbones. The maxilla is spongier than the mandible, due to the mandible’s dense cortical bone. Therefore, different bone defects require different bone substitute structures that match their structure and function. Producing structures with various architectures is relatively straightforward with DLP, known for its ability to print fine detail with high resolution. 3D printing also allows the fabrication of reproducible and consistent structures within multiple batches and can be based on patient-specific defects.

The structures themselves are produced by photopolymerisation, instead of relying on the material’s behaviour and cross-linking after the extrusion for extrusion-based printing or melting for fused deposition modelling (FDM). The bottom-up mechanism of DLP requires materials with the appropriate viscosity. If the viscosity is too low, the surface tension won’t be enough to allow the polymer to adhere to the printing platform and undergo base layer curing. If it is too viscous, it will not allow the uncured excess polymer to drain from the printed layer, reducing the printing resolution [10].

The viscosity is also dependent on the additive percentages. Incorporating additives into the photopolymer can improve its mechanical and biological properties. However, adding HA increased the mixture’s viscosity, which might interfere with the printing process. Additives such as HA can change the mixture’s rheological and optical properties. Additive particles can scatter the UV light and reduce the printing resolution [24]. Viscous polymers are usually harder to drain, particularly if their design involves micropores. The polymers will be trapped between the micropores and cured along with the subsequent layers. This will result in the loss of fine details, such as pores, which can play an essential role in cell biology.

The 3D printing results showed that the viscosity of the CSMA-2 and HA mixtures (0.3–0.5 Pa.s) was printable with the DLP method, and complex porous structures, such as the gyroid, could be printed. It has been reported that light-based 3D printing, such as stereolithography, requires viscosity under 5 Pa.s [25]. The CSMA-2 could also print in a 0.1 mm resolution setting, with a final 3D-printed resolution of approximately 0.08 mm. The difference between the printing resolution setting and the final 3D-printed resolution of the CSMA-2 resin might be caused by the high polymerisation due to the UV exposure during the curing process. This shrinkage effect on the 3D printing photopolymer is inevitable but can be minimised [26]. Since it is known that there was a 0.02 mm difference between the printing resolution and the 3D-printed layer thickness, the CAD can be adjusted by taking the difference into account. Therefore, CSMA-2 is suitable for light-based 3D printing with high accuracy and precision, based on its rheological properties and 3D printing results. It is important to find the balance between pre- and post-printing properties to ensure the printing-related properties will not be significantly affected while improving the 3D-printed structure properties.

The colour and opacity of the 3D-printed scaffolds were the only differences observed macroscopically between the groups. The colour is similar to that of human bone, therefore aesthetically acceptable as a biological implant. However, the SEM results demonstrated different surface morphologies of the scaffolds. The roughness of the surface increased following the increased percentage of HA. These results indicated that, although the printing process was not affected, the HA percentage of the polymer affects the surface morphology, particularly the surface roughness.

Since the surfaces of the 3D-printed scaffolds were different, the surface properties, such as the water contact angle and surface energy, were also different. The scaffold groups had a water contact angle of less than 80°, which indicates hydrophilicity (Figure 9a) [27]. The angle was higher on the CSMA-2 without the HA scaffolds. For the surface energy, it was the opposite. The CSMA-2 10HA 3D-printed structure showed the highest surface energy. Surface energy has been found to affect the hydrophilicity of a material surface. The higher the surface energy, the more hydrophilic the surface. Surface hydrophilicity can affect cell adhesion and proliferation, as well [28]. In addition, the surface energy on stiff materials has been reported to promote the osteogenic differentiation of stem cells [29]. From these results, the 3D-printed CSMA-2 scaffolds, with or without the HA, demonstrated hydrophilicity that can support cell proliferation and differentiation.

Adequate mechanical properties are also an essential factor for a successful bone implant. Bones are constantly exposed to mechanical loading, and bone substitutes should be able to withstand the force and surgical implantation procedure. The 3D-printed CSMA-2 porous gyroid scaffold had a compressive modulus of 0.4–0.5 N/mm^2^. These values were within the range of the human cancellous bone modulus with a porous or trabecular structure [30]. The ideal scaffolds for bone repair are expected to have a compressive strength comparable to that of native bone, and incorporating isosorbide has been reported to improve the mechanical properties of polymers [17,31]. The photoinitiator used in this work might also influence the mechanical properties. Previous studies have reported that BAPO was an efficient initiator for polymer cross-linking polymers such as poly(propylene fumarate) or PPF [32]. As mentioned before, the light absorption of BAPO is more suitable for most DLP printers with UV projectors that have a 405 nm wavelength. The match between the material and the 3D printer light source affects the mechanical properties of the 3D-printed structure. These findings indicated the suitability of CSMA-2 for high-resolution 3D printing that can fabricate non-toxic scaffolds with adequate strength.

To analyse the 3D-printed CSMA-2 scaffold’s cytocompatibility and ability to support osteogenic differentiation, hADSC were seeded. Figure 14 shows that the stem cells remained viable for up to 21 days on both media, with those in the growth medium showing higher metabolic activities. Polymers containing isosorbide, such as polyurethane, are known to support cell adhesion, proliferation, and differentiation [17]. Different metabolic activities between stem cells on the osteogenic and growth media might be caused by stem cells that were found to reduce their metabolic activity during differentiation [33]. Mature cells, such as osteocytes, slow the production of extracellular matrices that require high energy consumption. During the proliferation period, the progenitor cells show high glycolysis, whilst differentiation leads to decreasing glycolysis and increasing mitochondrial oxidation [34]. Low glycolysis has also been reported to decrease Alamar Blue reduction [35]. These findings could explain why the Alamar Blue reduction in the samples with the osteogenic medium was lower than in the samples with the growth medium. The hADSC were found to have lower metabolic activities on the scaffolds with the hydroxyapatite compared to the CSMA-2 0HA scaffolds. This result suggests the influence of hydroxyapatite on osteogenic differentiation since the faster the maturation process, the lower metabolic activity was found [33].

From Figure 8a–c, it can be seen that the CSMA-2 5HA and CSMA-2 10HA had rougher surfaces. Studies have reported that irregular surfaces can affect cell adhesion and morphologies. Scaffolds with a flat surface, smaller than the cell size, demonstrated elongated cell morphology and slower cell proliferation [36]. This can be caused by the lack of a surface that allows the cells to attach. Cells cultured on planar surfaces showed more mature adhesions compared to nano-grooved surfaces [37]. Similar reports also found that the adhesion and proliferation of cells on the surface with HA were slower than on smooth and flat culture plates [38]. These findings indicated that cell adhesion and proliferation are sensitive to the surface roughness that the HA addition affects.

Regarding differentiation, Figure 12 shows positive alizarin red staining on the scaffolds, indicating the secretion of calcium phosphate minerals by the hADSC. This result also suggests that the cells entered the mineralisation phase, a strong sign of osteogenic differentiation [39]. The staining on the scaffolds with HA was stronger than the ones without the HA, since the HA groups already contained calcium that could react with the staining. However, each scaffold group demonstrated the highest intensity on day 21, in line with the later stage of osteodifferentiation, where the matrix mineralisation occurs and calcium deposition increases [40].

The 3D-printed scaffold groups also showed a relatively similar expression of the osteogenic protein markers, which are OCN, OPN, and RUNX2. The expression of these proteins indicates the osteogenic differentiation of the cells on the scaffold from stem cells to mature osteoblasts or even osteocytes. OCN, or bone γ-carboxyglutamic acid (Gla), is a non-collagenous and the most abundant protein in the bone, which is only expressed by osteoblasts [41,42]. It is also regarded as a differentiation marker of the osteoblast [43]. Since calcium deposition is promoted in the presence of OCN, the OCN expression detected from day 7 (Figure 15) supported the positive result of alizarin red staining that indicates calcium deposition, which also can be observed from day 7 [40].

OPN, or secreted phosphoprotein 1 (SPP1), is a multifunctional protein involved in bone metabolism and remodelling. It is synthesised by osteoblasts, osteocytes, and other hematopoietic cells. The OPN gene expression of the hASDC was highest on day 7 of incubation on the CSMA-2 5HA scaffolds (Figure 16). The OPN gene expression decreased on day 14 and day 21 in every scaffold group. The hADSC on the CSMA-2 0HA showed the lowest OPN gene expression among the scaffold group. HA has been reported to induce the expression of osteo-specific genes on stem cells quite early, by influencing the material surface that leads to gene expression during the first few weeks of the incubation [44,45]. Cell adhesion to HA has been reported to induce signal transduction, leading to the sequential expression of genes involved in cell attachment, proliferation, and differentiation [46]. These gene expressions were caused by the Ca^2+^ ion release from the HA. [47] Ca^2+^ acts as a signalling messenger to induce osteogenic differentiation through BMPs/SMAD and RAS signalling pathways [48]. The result also suggested that the hADSC on the CSMA-2 0HA were still proliferating, whilst the other scaffolds groups underwent earlier proliferation arrest and started differentiating. Cells that are differentiating usually undergo proliferation arrest; this can explain the lower cell number and slower proliferation rate on the scaffolds with the HA (Figure 13) [49].

However, Figure 16 shows that the RUNX2 gene was expressed quite early by all the scaffold groups, including the CSMA-2 0HA, similar to the RUNX2 protein expression. RUNX2 is a protein essential for osteoblast differentiation and progenitor cell proliferation [50]. RUNX2 is required for preosteoblast proliferation and inducing the commitment of stem cells to differentiate into osteoblast lineage cells [51]. Since RUNX2 is weakly expressed in uncommitted mesenchymal stem cells, the expression of RUNX2 in the adipose-derived stem cells on the CSMA-2 scaffolds indicated their differentiation to immature osteoblasts [51]. Different from those of the OPN and OCN, the RUNX2 gene expressions in our result were not affected by the HA percentage on the scaffold. The presence of aliphatic side chains and cyclohexenes on the CSMA-2 that increased the surface charge of the 3D-printed scaffold might be able to promote differentiation without the help of HA [52]. Furthermore, the metabolic activity was significantly lower in the CSMA-2 scaffolds incubated with the osteogenic medium (Figure 10). This can be caused by the RUNX2 expression that arrests cells in the G0/G1 phase and activates expressions of other genes related to osteogenic differentiation [53].

Since CSMA-2 scaffolds can support stem cell differentiation without adding growth factors or protein, the application will be more straightforward. In a clinical application, an osteogenic scaffold can help the surrounding progenitor cells from the periosteum or the native bone to differentiate into bone cells and initiate bone repair [54]. When combined with stem cells as a regenerative medicine approach, a 3D-printed osteogenic scaffold can also enable the differentiation process of the incorporated stem cells. Thus, the scaffolds promoting osteogenic differentiation have more advantages for patients.

## 5. Conclusions

This study describes the development and optimisation of CSMA-2, an isosorbide-based polymer that showed compatibility with the DLP 3D printing method. The DLP method allows the fabrication of structures with high resolution, accuracy, and precision compared to the commonly used extrusion-based 3D printing. Complex gyroid scaffolds with interconnected pores were successfully printed and demonstrated good mechanical properties, similar to those of human cancellous bone. The 3D-printed scaffolds also supported cell proliferation and promoted osteogenic differentiation, indicating promising applications in bone tissue engineering. Future work will be focused on optimising the 3D printing parameters and cell seeding methods to fabricate better 3D-printed patient-specific implants.

## Figures and Tables

**Figure 1 biomolecules-12-01692-f001:**
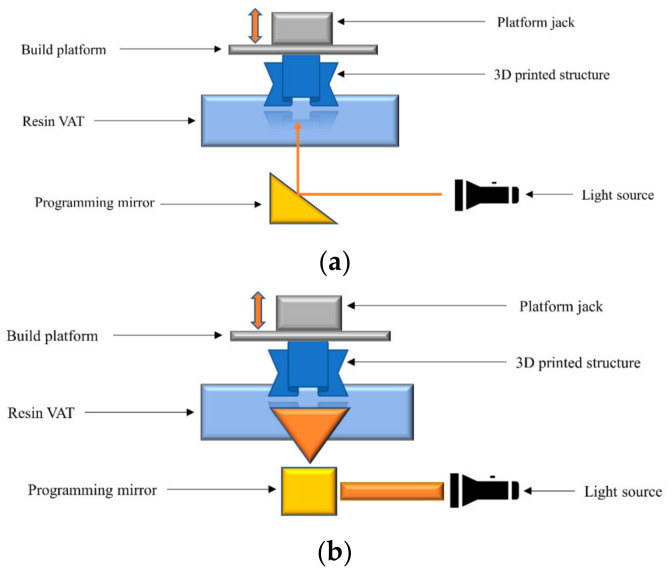
Schematic diagram of SLA 3D (**a**) Schematic diagram of DLP (**b**) Reproduced from [11].

**Figure 2 biomolecules-12-01692-f002:**
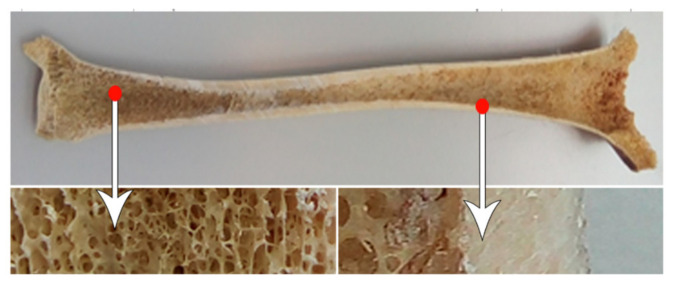
Structure of the human bone. Reproduced from [13].

**Figure 3 biomolecules-12-01692-f003:**
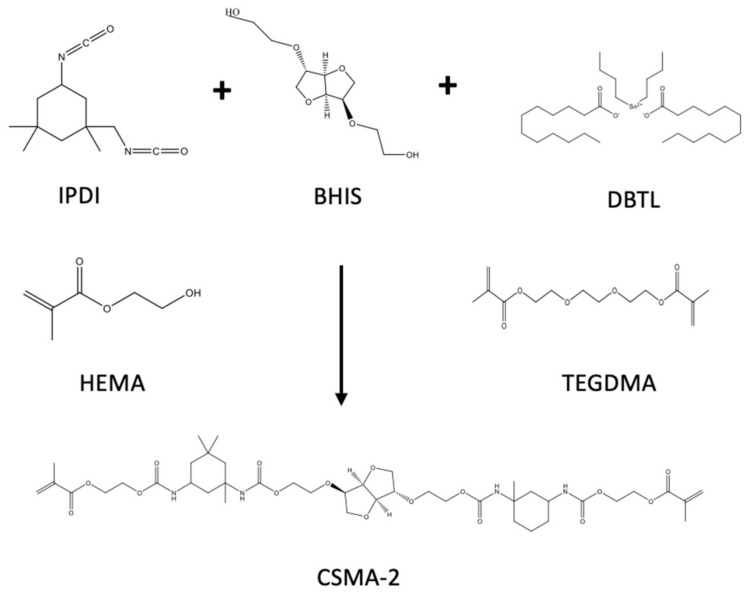
Schematic 2-step reactions of CSMA-2 synthesis.

**Figure 4 biomolecules-12-01692-f004:**
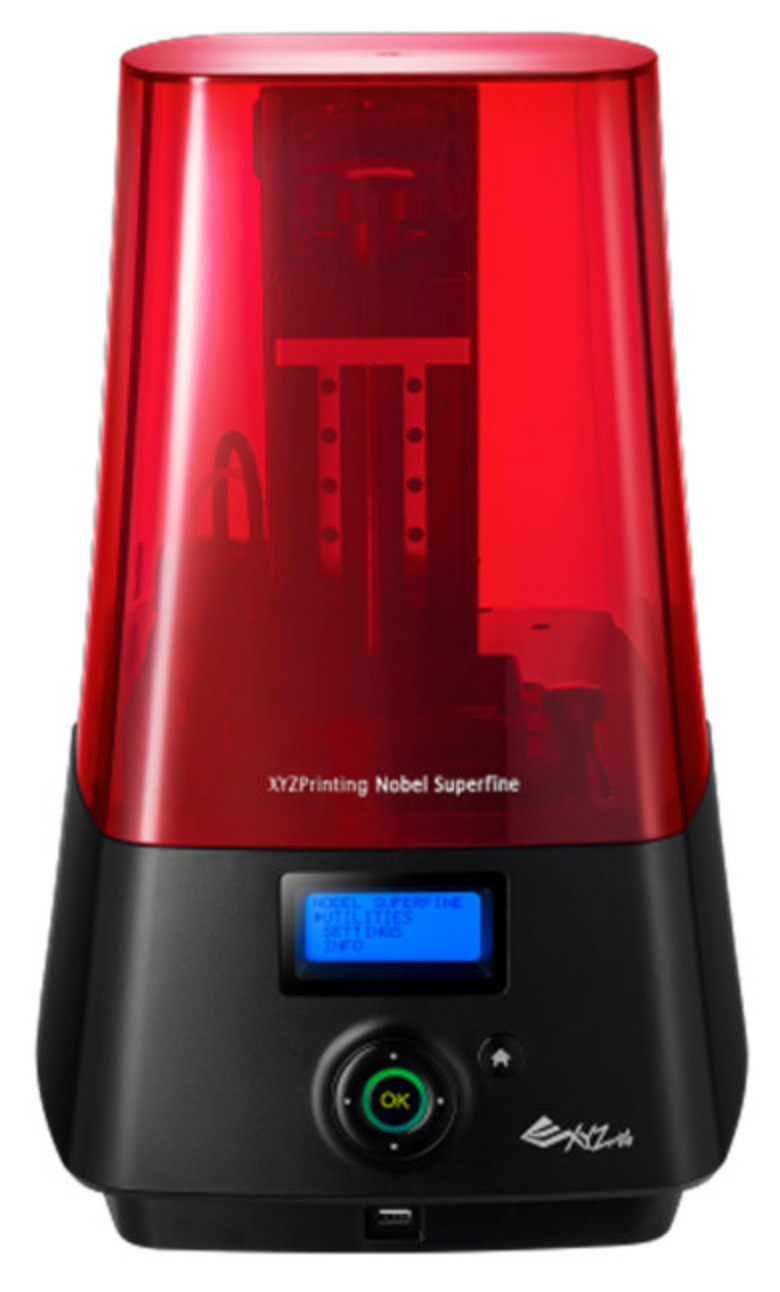
DLP 3D Printer.

**Figure 5 biomolecules-12-01692-f005:**
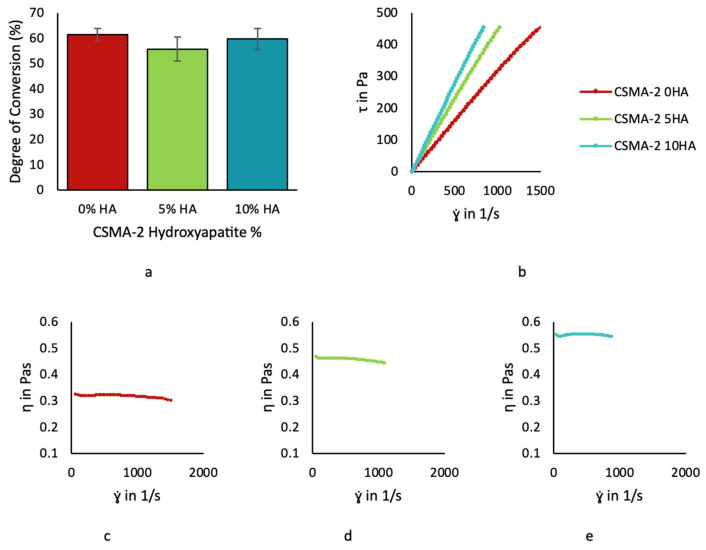
CSMA-2 monomer degree of conversion (**a**). Linear stress-stain relationship of CSMA-2 monomer (**b**). Constant viscosity throughout different shear strain of CSMA-2 0HA (**c**). CSMA-2 5HA (**d**). and CSMA-2 10HA (**e**).

**Figure 6 biomolecules-12-01692-f006:**
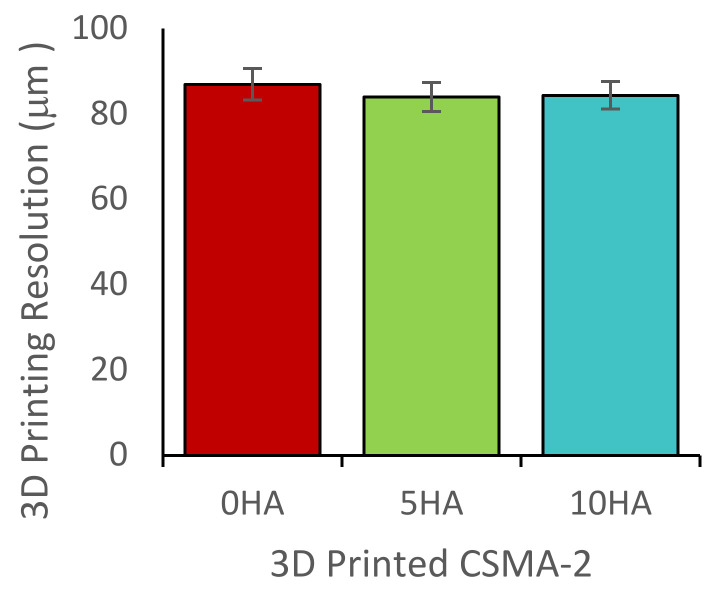
Final resolution of 3D-Printed CSMA-2 structure.

**Figure 7 biomolecules-12-01692-f007:**
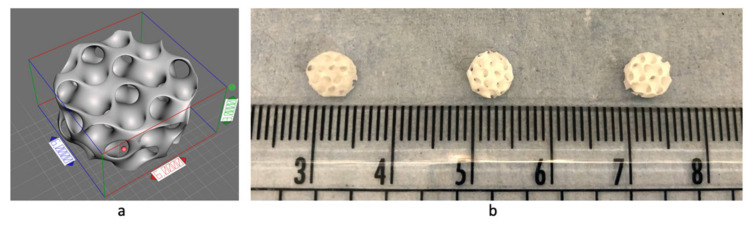
CAD of gyroid scaffold (**a**). 3D-printed structures of the CAD with different CSMA-2 and HA formulations, from left to right: 0HA, 5HsA, and 10HA (**b**).

**Figure 8 biomolecules-12-01692-f008:**
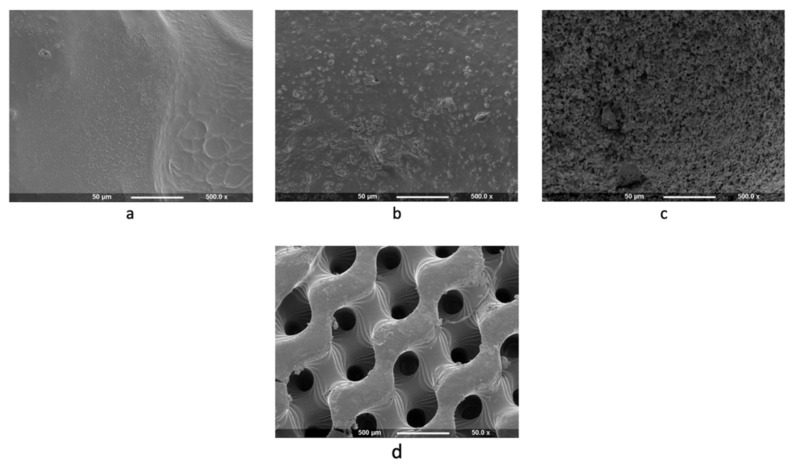
SEM Images show different surface morphology of 3D-printed gyroid scaffold, CSMA-2 0HA (**a**), CSMA-2 5HA (**b**), and CSMA-2 10HA (**c**). 3D-printed gyroid structures are visible (**d**).

**Figure 9 biomolecules-12-01692-f009:**
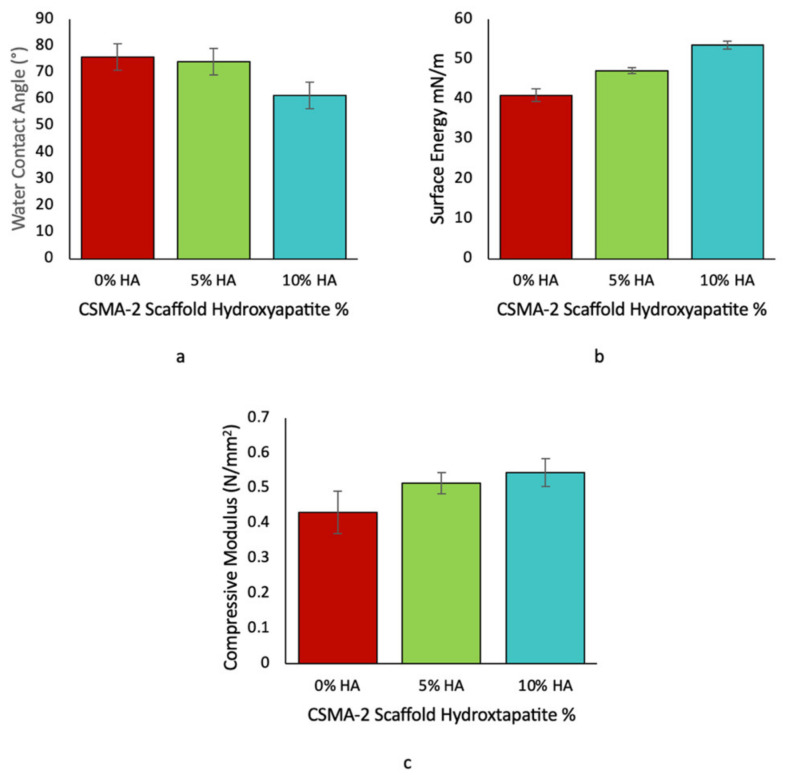
The water contact angle of 3D-printed CSMA-2 scaffolds (**a**). The surface energy of 3D-printed CSMA-2 scaffolds (**b**). The compressive modulus of 3D-printed CSMA-2 scaffolds (**c**).

**Figure 10 biomolecules-12-01692-f010:**
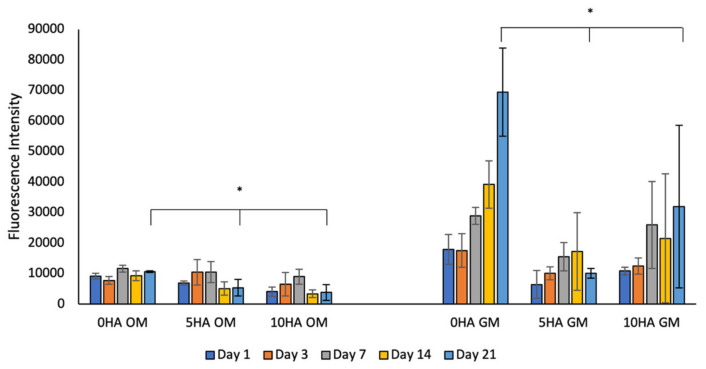
Metabolic activity result of hADSC cells seeded on 3D-printed CSMA-2 scaffolds with different media. OM: osteogenic medium, GM: growth medium. * *p* < 0.05.

**Figure 11 biomolecules-12-01692-f011:**
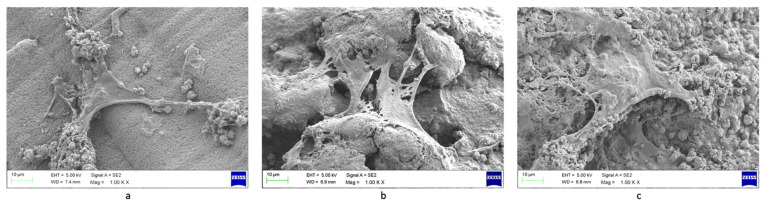
SEM images of hADSC attachment on 3D-printed CSMA-2 scaffolds. (**a**) CSMA-2-0HA (**b**) CSMA-2 5HA (**c**) CSMA-2 10HA.

**Figure 12 biomolecules-12-01692-f012:**
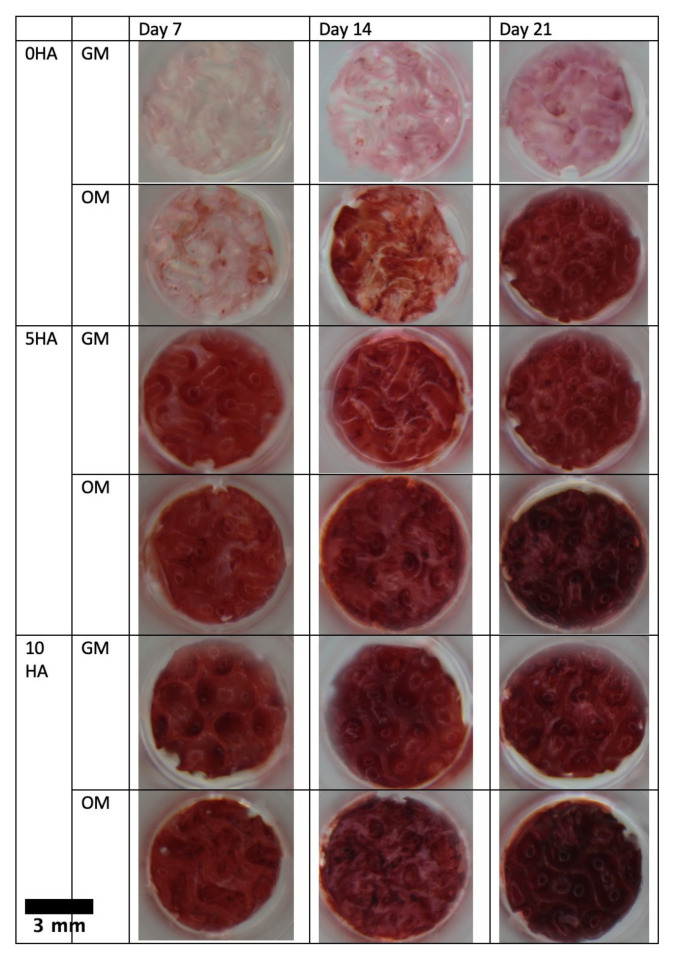
Alizarin red staining results of hADSC incubated on CSMA-2 scaffold.

**Figure 13 biomolecules-12-01692-f013:**
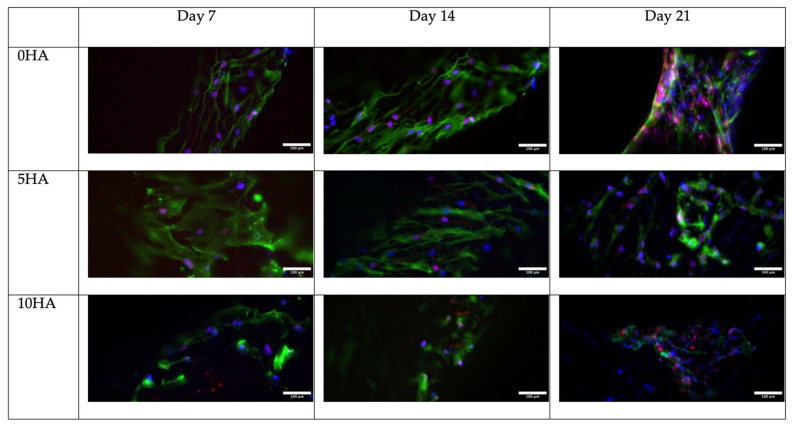
Immunofluorescence images of RUNX2 staining (red), DAPI staining on nuclei (blue), and Phalloidin on F-Acting (green) in hADSC cultured on 3D-printed CSMA-2 scaffolds with different HA percentages. Scale bars: 100 μm.

**Figure 14 biomolecules-12-01692-f014:**
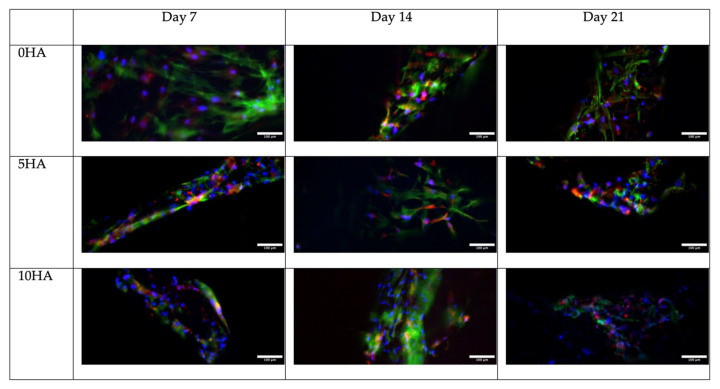
Immunofluorescence images of OPN staining (red), DAPI staining on nuclei (blue), and Phalloidin on F-Actin (green) in hADSC cultured on 3D-printed CSMA-2 scaffolds with different HA percentages. Scale bars: 100 μm.

**Figure 15 biomolecules-12-01692-f015:**
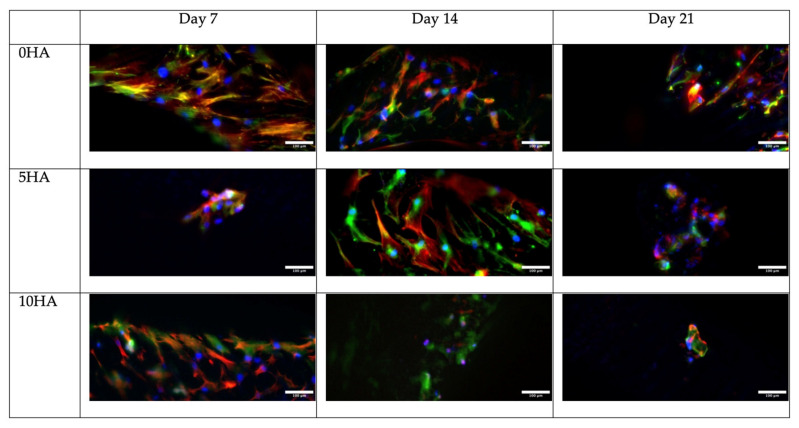
Immunofluorescence images of OCN staining (green), DAPI staining on nuclei (blue), and Phalloidin on F-Actin (red) in hADSC cultured on 3D-printed CSMA-2 scaffolds with different HA percentages. Scale bars: 100 μm.

**Figure 16 biomolecules-12-01692-f016:**
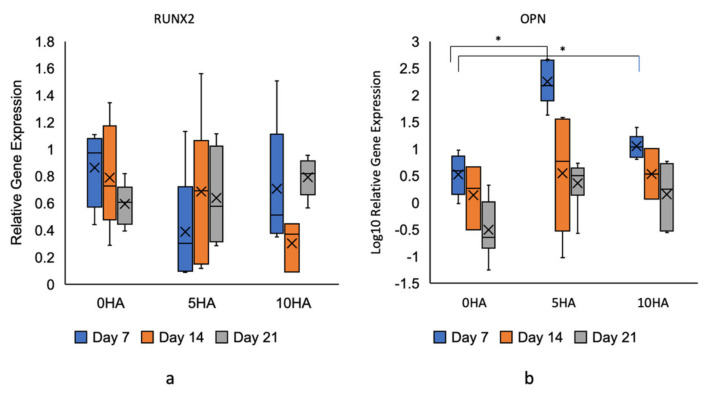
Gene expression of hADSC seeded on 3D-printed CSMA-2 scaffolds. (**a**) RUNX2 gene expression, (**b**) OPN gene expression. * *p* < 0.05.

## Data Availability

Data will be made available upon request to the corresponding author.

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
