# Peer review of "Digital Light Processing 3D Printing of Gyroid Scaffold with Isosorbide-Based Photopolymer for Bone Tissue Engineering"

_biomolecules, 2022, doi:10.3390/biom12111692_

Round 1

Reviewer 1 Report

please see attached pdf

Author Response

We have edited ALL the points as requested. Our edits are marked in the reply to the reviewer document in RED and any edits in the manuscript are also in red for ease of finding.

Reviewer 2 Report

The manuscript from Fiona Verisqa et al. designed CSMA-2, a photopolymer based on Isosorbide, were used to evaluate as bone substitute for regeneration using DLP printing.

This manuscript describes the physicochemical and biological properties of 3D Printing of Gyroid Scaffold. In this study, 3D printed CSMA-2 porous gyroid scaffold containing with different HA% have adequate compressive modulus and good biocompatibility and to improve hADSC proliferation and osteogenesis process. Although it is not extensive in some aspects and some pieces of data are missing, it involves a potential material possibility that allow the 3D printing fabrication of complex biomimetic scaffolds for bone regeneration. Thus, in my opinion, this work could perhaps be considered at “Biomolecules” after minor revisions.

Minor:

1.        We are curious about the adhesion of hADSC on 3D printed CSMA-2 porous gyroid scaffold. Could authors provide SEM images that could show the morphology/ topography images for more detail?

2.        Based on Fig.11, the result is not enough to support the value of significant different. Please add detailed description for the different between 3 group with various periods.

3.        The qualitative and quantitative analysis of RUNX2, OPN and OCN show in Fig. 8-12. According to HA can interact with the cells and generate potential inductive to differentiate into the osteolineage. Could authors describe for more detail the correlation?

4.        The development, formation, and homeostasis of bone tissue involve mesenchymal cell lineages in the synthesis, deposition, and remodeling of type I collagen. Have authors consider to analyze the expression of synthesis of type I collagen?

Author Response

We have answered ALL four comments made by the referee with extra data and also added extra text into the manuscript (highlighted in red for ease of finding.

Round 2

Reviewer 1 Report

I want to thank the authors for their reviews and targeted interventions. The manuscript looks much better now, I consent to its publication.